# MULTI-OBJECTIVE NEURAL ARCHITECTURE SEARCH VIA PREDICTIVE NETWORK PERFORMANCE OPTIMIZATION

## ABSTRACT

Neural Architecture Search (NAS) has shown great potentials in finding a better neural network design than human design. Sample-based NAS is the most fundamental method aiming at exploring the search space and evaluating the most promising architecture. However, few works have focused on improving the sampling efficiency for a multi-objective NAS. Inspired by the nature of the graph structure of a neural network, we propose BOGCN-NAS, a NAS algorithm using Bayesian Optimization with Graph Convolutional Network (GCN) predictor. Specifically, we apply GCN as a surrogate model to adaptively discover and incorporate nodes structure to approximate the performance of the architecture. For NAS-oriented tasks, we also design a weighted loss focusing on architectures with high performance. Our method further considers an efficient multi-objective search which can be flexibly injected into any sample-based NAS pipelines to efficiently find the best speed/accuracy trade-off. Extensive experiments are conducted to verify the effectiveness of our method over many competing methods, e.g. $128.4\times$ more efficient than Random Search and $7.8\times$ more efficient than previous SOTA LaNAS for finding the best architecture on the largest NAS dataset NASBench-101.

## 1 INTRODUCTION

Recently Neural Architecture Search (NAS) has aroused a surge of interest by its potentials of freeing the researchers from tedious and time-consuming architecture tuning for each new task and dataset. Specifically, NAS has already shown some competitive results comparing with hand-crafted architectures in computer vision: classification (Real et al., 2019b), detection, segmentation (Ghiasi et al., 2019; Chen et al., 2019; Liu et al., 2019a) and super-resolution (Chu et al., 2019). Meanwhile, NAS has also achieved remarkable results in natural language processing tasks (Luong et al., 2018; So et al., 2019).

A variety of search strategies have been proposed, which may be categorized into two groups: one-shot NAS algorithms (Liu et al., 2019b; Pham et al., 2018; Luo et al., 2018), and sample-based algorithms (Zoph & Le, 2017; Liu et al., 2018a; Real et al., 2019b). One-shot NAS algorithms embed the architecture searching process into the training stage by using weight sharing, continuous relaxation or network morphisms. However, those methods cannot guarantee the optimal performance of the final model due to those approximation tricks and is usually sensitive to the initial seeds (Sciuto et al., 2019). On the other hand, sample-based algorithms are relatively slower but reliable. They explore and exploit the search space using some general search algorithms by providing potential candidates with higher accuracy. However, it requires fully training of huge amounts of candidate models.

Typically, the focus of most existing NAS methods has been on the accuracy of the final searched model alone, ignoring the cost spent in the search phase. Thus, the comparison between existing search algorithms for NAS is very difficult. (Wang et al., 2019b) gives us an example of evaluating the NAS algorithms from this view. They compare the number of training architectures sampled until finding the global optimal architecture with the top accuracy in the NAS datasets. Besides accuracy, in real applications, there are many other objectives we should concern, such as speed/accuracy

trade-off. Hence, in this paper, we aim at designing an efficient multi-objective NAS algorithm to adaptively explore the search space and capture the structural information of architectures related to the performance.

The common issue faced by this problem is that optimizing objective functions is computationally expensive and the search space always contains billions of architectures. To tackle this problem, we present BOGCN-NAS, a NAS algorithm that utilizes Bayesian Optimization (BO) together with Graph Convolutional Network (GCN). BO is an efficient algorithm for finding the global optimum of costly black-box function (Mockus et al., 1978). In our method, we replace the popular Gaussian Processes model with a proposed GCN model as the surrogate function for BO (Jones, 2001). We have found that GCN can generalize fairly well with just a few architecture-accuracy pairs as its training set. As BO balances exploration and exploitation during searching and GCN extracts embeddings that can well represent model architectures, BOGCN-NAS is able to obtain the optimal model architecture with only a few samples from the search space. Thus, our method is more resource-efficient than the previous ones. Graph neural network has been proposed in previous work for predicting the parameters of the architecture using a graph hypernetwork (Zhang et al., 2019). However, it's still a one-shot NAS method and thus cannot ensure the performance of the final found model. In contrast, we use graph embedding to predict the performance directly and can guarantee performance as well.

The proposed BOGCN-NAS outperforms current state-of-the-art searching methods, including Evolution (Real et al., 2019b), MCTS (Wang et al., 2019b), LaNAS (Wang et al., 2019a). We observe consistent gains on multiple search space for CV and NLP tasks, i.e., NASBench-101 (denoted NAS-Bench) (Ying et al., 2019) and LSTM-12K (toy dataset). In particular, our method BOGCN-NAS is $128.4\times$ more efficient than Random Search and $7.8\times$ more efficient than previous SOTA LaNAS on NASBench (Wang et al., 2019a). We apply our method to multi-objective NAS further, considering adding more search objectives including accuracy and number of parameters. Our method can find more superior Pareto front on NASBench. Our algorithm is applied on open domain search with NASNet search space and ResNet Style search space, which finds competitive models in both scenarios. The results of experiment demonstrate our proposed algorithm can find a more competitive Pareto front compared with other sample-based methods.

## 2 RELATED WORK

### 2.1 BAYESIAN OPTIMIZATION

Bayesian Optimization aims to find the global optimal over a compact subset $\mathcal{X}$ (here we consider maximization problem):

$$x^* = \arg\max_{x \in \mathcal{X}} f(x). \tag{1}$$

Bayesian Optimization considers prior belief about objective function and updates posterior probability with online sampling. Gaussian Processes (GPs) is widely used as a surrogate model to approximate the objective function (Jones, 2001). And *Expected Improvement* acquisition function is often adopted (Mockus et al., 1978). For the hyperparameters of the surrogate model $\Theta$, we define

$$\gamma(x) = \frac{\mu(x; \mathcal{D}, \Theta) - f(x_{best})}{\sigma(x; \mathcal{D}, \Theta)}, \tag{2}$$

where $\mu(x; \mathcal{D}, \Theta)$ is the predictive mean, $\sigma^2(x; \mathcal{D}, \Theta)$ is the predictive variance and $f(x_{best})$ is the maximal value observed. The *Expected Improvement* (EI) criterion is defined as follows.

$$a_{EI}(x; \mathcal{D}, \Theta) = \sigma(x; \mathcal{D}, \Theta)[\gamma(x)\Phi(\gamma(x); 0, 1) + \mathcal{N}(\gamma(x); 0, 1)], \tag{3}$$

where $\mathcal{N}(\cdot; 0, 1)$ is the probability density function of a standard normal and $\Phi(\cdot; 0, 1)$ is its cumulative distribution.

### 2.2 MULTI-OBJECTIVE OPTIMIZATION

Without loss of generality about $max$ or $min$, given a search space $\mathcal{X}$ and $m \geq 1$ objectives $f_1 : \mathcal{X} \to \mathbb{R}, \ldots, f_m : \mathcal{X} \to \mathbb{R}$, variable $X_1 \in \mathcal{X}$ dominates variable $X_2 \in \mathcal{X}$ (denoted $X_1 \succ X_2$) if (i) $f_i(X_1) \geq f_i(X_2), \forall i \in \{1, \ldots, m\}$; and (ii) $f_j(X_1) > f_j(X_2)$ for at least one $j \in \{1, \ldots, m\}$. $X^*$

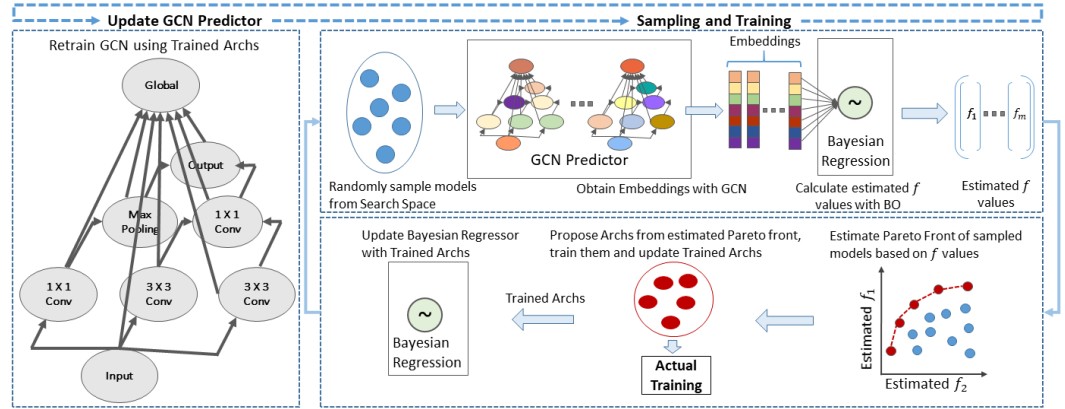

Figure 1: The overview of our proposed algorithm. BOGCN-NAS is integrated by GCN and Bayesian Linear Regression, including two iterative phases: 1) Sampling and Training; 2) Update GCN predictor. During phase-one, we randomly sample a group of architectures from the search space as candidate pool and calculate a predicted objective $f$. Then an estimated Pareto front is constructed by $f$ and Pareto optimal points are selected as proposed architectures. Then we fully train them and add into trained architecture set ($U$). The Bayesian regression is updated with $U$. During phase-two, we further retrain the GCN predictor with $U$. When there is only one objective, the Pareto front reduces to one optimal architecture.

is *Pareto optimal* if there is no $X \in \mathcal{X}$ that domaines $X^*$. The set of all Pareto optimal architectures consitutes the *Pareto front* $\mathcal{P}_f$. A multi-objective optimization problem (MOP) aims at finding such input $X \in \mathcal{X}$ that $X$ cannot be dominated by any variable in $\mathcal{X}$ (Marler & Arora, 2004).

## 2.3 GRAPH CONVOLUTIONAL NETWORK

Let the graph be $G = (V, E)$, where $V$ is a set of $N$ nodes, and $E$ is the set of edges. Let its adjacency matrix be $A$ and feature matrix be $X$. The graph convolutional network (GCN) is a learning model for graph-structure data (Kipf & Welling, 2016). For a $L$-layer GCN, the layer-wise propagation rule is given by:

$$H^{(l+1)} = f(H^{(l)}, A) = \text{ReLU}(\widetilde{D}^{\frac{1}{2}} \widetilde{A} \widetilde{D}^{-\frac{1}{2}} H^{(l)} W^{(l)}), \qquad (4)$$

where $\widetilde{A} = A + I$, $I$ is the identity matrix, $\widetilde{D}$ is a diagonal matrix with $\widetilde{D}_{ii} = \sum_{j=1}^{N} A_{ij}$, $H^{(l)}$ and $W^{(l)}$ are the feature map and weight at the $l$-th layer respectively, and $\text{ReLU}(\cdot)$ is the ReLU activation function. $H^{(0)}$ is the original feature matrix $X$, and $H^{(L)}$ is the graph embedding matrix.

## 3 BOGCN-NAS

To search for the optimal architecture more efficiently, we proposed BOGCN-NAS by using predictive network performance Optimization with the GCN (Section 3.2) while utilizing the Bayesian Optimization. Figure 1 shows the overview of the proposed algorithm.

## 3.1 MULTI-OBJECTIVE NAS

We formulate NAS problem as a multi-objective optimization problem over the architecture search space $\mathcal{A}$ where objective functions can be accuracy, latency, number of parameters, etc. We aim to find architectures on the Pareto front of $\mathcal{A}$. Specifically, when $m = 1$, it reduces to single-objective (usually accuracy) NAS and the corresponding Pareto front reduces to one optimal architecture.

## 3.2 GCN PREDICTOR

GCN predictor predicts the performance (like accuracy) of an architecture. Compared with MLP and LSTM predictors proposed before (Wang et al., 2019b), GCN can preserve the context of graph data better. Another important characteristic of GCN is its ability to handle variable number of nodes, while an MLP cannot take a larger architecture as the input. Even though the LSTM can handle variable-length sequences, its performance is not competitive because of the flat string encoding.

A neural network can be viewed as a directed attributed graph, in which each node represents an operation (such as convolution operation) and each edge represents a data flow. As a concrete illustration, we use the architectures in the NASBench dataset (Ying et al., 2019) as an example. The idea can be easily extended to other architectures. In NASBench, each architecture is constituted by stacking multiple repeated cells. Thus, we will focus on searching the cell architecture. An example cell in NASBench is illustrated on the left side of Figure 1, where "input" represents the input of the cell, "output" represents the output of the cell, "$1 \times 1$ Conv, $3 \times 3$ Conv, Max Pooling" are three different operations (5 operations totally).

We propose to encode the cell into an adjacency matrix $A$ (asymmetric) and a feature matrix $X$, as the input of our GCN predictor. Note that the vanilla GCN only extracts the node embeddings, while we want to obtain graph embedding. Following (Scarselli et al., 2008), we add a global node to the original graph of the cell and let every node point at the global node. The adjacency matrix can be obtained directly from the graph structure. For the feature matrix, we use the one-hot coding scheme for each operation. Besides the original 5 different operations defined in NASBench, we add another operation (global node) into coding scheme.

We feed $A$ and $X$ to a multi-layer GCN model to obtain the embedding of every node $H^{(L)}$ by (Eq. 4). For high-level prediction, we leave original nodes out and take the embedding of global node solely because it already has the overall context of the architecture. Followed by one fully-connected layer with sigmoid activation function, we can get the predicted accuracy. In training phase, we use MSE loss for regression optimization.

## 3.3 INCORPORATING BO INTO GCN

Bayesian Optimization is an efficient model for search and optimization problems, which considers balancing both exploitation and exploration balanced. It depends on updating the posterior distribution with the samples drawn from the search space based on one cheap surrogate model.

GPs is one of the most popular choices because Gaussian distribution is self-conjugate such that a posterior distribution is also the same form as the prior. (Kandasamy et al., 2018) and (Jin et al., 2019) both define the heuristic distance between neural architectures and use GPs with defined distence for searching. However, since the computation increases cubically with the number of samples (Snoek et al., 2015), GPs is so costly for NAS problem, whose search space is always huge. Another drawback for GPs is that it cannot handle graph-data directly without a special encoding scheme. In this work, we replace the popular surrogate model with our GCN predictor and take the uncertainty into consideration.

Inspired by previous work (Snoek et al., 2015), we train the GCN predictor first with the trained architecture set $\mathcal{D}$ containing architectures $\{(A_i, X_i)\}_{i=1}^n$ with their actual performances $\{t_i\}_{i=1,}^n$ then during searching, we replace the last fully connected layer with Bayesian Linear Regressor (BLR) for Bayesian estimation and retain GCN related layers for point estimation. We only consider the uncertainty of the weights on the last fully-connected layer. We denote the embedding function of global node by $\phi(\cdot, \cdot) = [\phi_1(\cdot, \cdot), \phi_2(\cdot, \cdot), ..., \phi_d(\cdot, \cdot)]^T$. We can get the embedding of every architecture $\phi(A_i, X_i)$ from the trained architecture set $\mathcal{D}$ and treat them as the basis functions for BO. For clarifying, we define $\Phi$ to be the design matrix where $\Phi_{ij} = \phi_j(A_i, X_i)$.

Different from typical Bayesian Linear Regression (BLR) (Bishop, 2006), the final layer of our GCN predictor contains non-linear activation function. Here we use the inverse function trick to avoid the non-trivial variant. Without fitting the true accuracy $t$, we prefer to estimate the value before the activation function $logit(t)$ such that we can convert non-linear regression to linear regression problem. The key of BO is the order of the acquisition function over all sampled architectures rather than true values. Due to the monotonicity of sigmoid function, the order property still holds.

The predicted mean and variance given by our model without the last activation function are shown below.

$$\mu(A, X; \mathcal{D}, \alpha, \beta) = m_N^T \phi(A, X), \tag{5}$$

$$\sigma^2(A, X; \mathcal{D}, \alpha, \beta) = \frac{1}{\beta} + \phi(A, X)^T S_N \phi(A, X), \tag{6}$$

where

$$m_N = \beta S_N \Phi^T logit(t), S_N = \alpha I + \beta \Phi^T \Phi. \tag{7}$$

Here, $\alpha, \beta$ are precisions of the prior, which are hyperparameters of the Bayesian Optimization model. We can estimate them by maximizing the log marginal likelihood as following (Snoek et al., 2012).

$$\log p(logit(t)|\alpha, \beta) = \frac{M}{2} \log \alpha + \frac{N}{2} \log \beta - \frac{\beta}{2} ||logit(t) - \Phi m_N||^2$$

$$- \frac{\alpha}{2} m_N^T m_N - \frac{1}{2} \log |S_N^{-1}| - \frac{N}{2} \log 2\pi. \tag{8}$$

### 3.4 SEARCH WITH ALTERNATE LEARNING

NAS is a process of online learning during which we can utilize new fully-trained architectures sampled from the search space. Therefore, the GCN model and BLR in BOGCN-NAS should be updated with the increasing number of samples for better generalization. In this work, for the reason that GCN retraining is more expensive than BLR updating, we update BLR more frequently than GCN predictor.

The algorithm of our proposed BOGCN-NAS is illustrated in Algorithm 1. Given the search space $\mathcal{A}$, we initialize trained architecture sets $U$ containing architectures $(A_i, X_i)$ with their performance $t_i = \{f_{1i}, \ldots, f_{mi}\}$. We train GCN predictor firstly with $U$ and replace the last fully-connected layer with BLR described in Section 3.3. Then we randomly sample a subspace $\mathcal{R}$ as following candidate pool if $\mathcal{A}$ is such huge that we cannot cover every architecture. After obtaining the candidate pool, we can calculate every candidate model's *Expected Improvement* as their estimated objective values $\hat{t}_j = \{\hat{f_{1j}}, \ldots, \hat{f_{mj}}\}$. Based on $\hat{t}_j$ and multi-objective formulation (Section 3.1), we can generate a estimated Pareto Front and sample estimated Pareto optimal models as set $\mathcal{S}$ and fully-train them to obtain the true objective values $t_j$. The trained architecture sets is then updated by $U = U \cup S$. Accumulating a certain amount of new data, we update GCN and BLR alternately for next periodic sampling until satisfying the stop criteria.

### 3.5 EXPONENTIAL WEIGHTED LOSS

For GCN predictor alone, MSE loss can achieve competitive performance for fitting all data. However, when it comes to the surrogate function for finding the top-performance model, we should pay more attention to architectures with high performance compared to others.

In the search phase of BO, we select samples with top values of acquisition function. Specifically, we expect to predict architectures with high performance as accurately as possible, while distinguishing low-performance architecture is sufficient for our purpose. For adapting NAS task, we propose Exponential Weighted Loss for the surrogate GCN model and replace common MSE loss with weighted loss completely in Algorithm 1.

$$L_{exp} = \frac{1}{N(e-1)} \sum_{i=1}^{N} (\exp(\widetilde{y}_i) - 1)||y_i - \widetilde{y}_i||^2, \tag{9}$$

---

**Algorithm 1** BOGCN-NAS Search Procedure. $\mathcal{A}$ is the given search space, $l$ is the number of samples every iteration, $k$ is the ratio of $GCN/BLR$ update times, $\mathcal{P}_f$ is the optimal Pareto front and $threshold$ is the criteria for search stopping.

---

1: Initialize trained architecture sets $U = \{A_i, X_i, t_i\}_{i=1}^n$ from $\mathcal{A}$ and current Pareto front $\widetilde{\mathcal{P}_f}$;
2: Train $GCN$ and $BLR$ initially;
3: **while** $|\widetilde{\mathcal{P}_f} \cap \mathcal{P}_f|/|\mathcal{P}_f| < threshold$ **do**
4:     **for** iteration= $1, 2, ..., k$ **do**
5:         $\mathcal{R} = random\_sample(\mathcal{A})$;
6:         $S = sample(GCN, BLR, l, \mathcal{R})$;
7:         Fully train sampled models $S$;
8:         Update $U = U \cup S$;
9:         Update $\widetilde{\mathcal{P}_f}$ in set $U$;
10:        $BLR.update(GCN, U)$;
11:     **end for**
12:     $GCN.retrain(U)$;
13: **end while**
    return $\mathcal{P}_f$

**function** $sample(GCN, BLR, l, \mathcal{R})$
15:    $embeddings = GCN(\mathcal{A})$;
16:    Predict mean and variance of models in $\mathcal{R}$ with $BLR$ and $embeddings$ by (Eq. 5) and (Eq. 6);
17:    Calculate corresponding Expected Improvement by (Eq. 3);
18:    Sample top $l$ Pareto optimal models sorted by Expected Improvement into set $S$;
19:    **return** $S$.
20: **end function**

---

here $y_i$ is the predicted accuracy, $\widetilde{y_i}$ is the ground truth and $e - 1$ is normalization factor ($e$ is the base of the natural logarithm). Thus, our predictor will focus on predicting for those models with higher accuracy.

## 4 EXPERIMENT

**Dataset and Search Space** In this section, we validate the performance of the proposed BOGCN-NAS on NASBench (Ying et al., 2019). This is the largest benchmark NAS dataset for computer vision task currently, with $420K$ architectures. To show the generalization of our method, we collect $12K$ LSTM models trained on PTB dataset (MARCUS et al., 1993) aiming at NLP task. We also compare our method in open domains with NASNet search space (Zoph et al., 2018) and ResNet style search space (Li et al., 2019). We will provide the detail of the search space and collection method in Appendices Section.

Specifically, we prove the performance of single proposed GCN Predictor firstly and then validate the whole framework on single-objective/multi-objective search problems compared with other baselines (with default settings). The results of the experiment illustrate the efficiency of BOGCN-NAS.

### 4.1 COMPARISON OF INDIVIDUAL PREDICTORS

We use a four-layer GCN, with 64 hidden units in each layer, as the surrogate model for BO. During training, we use the Adam optimizer (Kingma & Ba, 2014), with a learning rate of 0.001 and a mini-batch size of 128. The proposed GCN predictor is compared with an MLP predictor and LSTM predictor (Liu et al., 2018a; Wang et al., 2019b). Here we apply MSE loss because we compare the stand-alone predictors, while we replace with Exponential Weighted Loss for subsequent NAS problems. For the evaluation metric, we follow (Wang et al., 2019b) and use the correlation between the predicted accuracy and true accuracy. One difference between our evaluation method and (Wang et al., 2019b) is that we compare predictor's performance training on fewer architectures rather than whole search space, because NAS start with little training data and we cannot train such a large number of architectures in practice. We used 1000 architectures in NASBench for training, 100 architectures for validation, and 10000 architectures for testing.

Table 1 shows the correlation result of various predictors and the number of predictors' parameters. The prediction result of three predictors is also illustrated in Figure 2 and the value means the

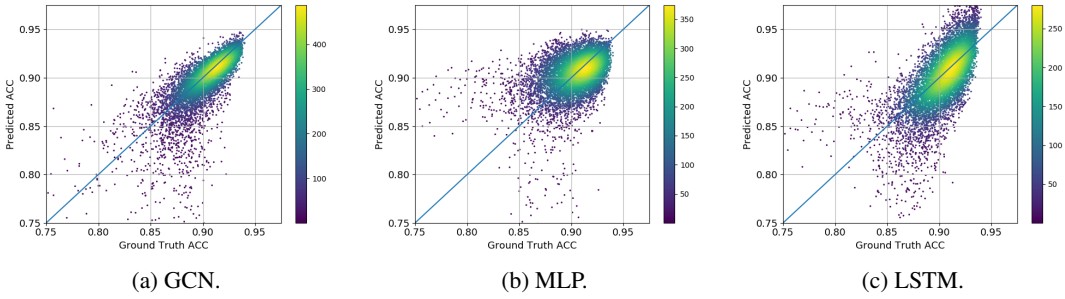

(a) GCN.                          (b) MLP.                          (c) LSTM.

Figure 2: Correlation of predicted accuracy versus ground truth for various predictors on NASBench. It can be found that our GCN predictor has better correlation performance than the competing predictors.

Gaussian kernel-density estimation. As can be seen, the GCN predictor can predict the performance of architectures more accurately than the other two predictors. In addition, the GCN predictor has fewer parameters compared with them.

Table 1: Performance of predictors training on small dataset. Our method has much better correlation with less predictor parameters.

| | Correlation | No. of parameters |
|---|---|---|
| MLP | 0.400 | 6326K |
| LSTM | 0.460 | 92K |
| GCN | **0.607** | **14K** |

Table 2: Comparison of samples required to find the global optimal over 50 rounds.

| | NASBench | LSTM-12K |
|---|---|---|
| Random | 188139.8 | 6182.6 |
| Reg Evolution | 87402.7 | 5670.9 |
| MCTS | 73977.2 | 4687.4 |
| LaNAS | 11390.7 | 2464.4 |
| BOGCN-NAS | **1465.4** | **558.8** |

## 4.2 SINGLE-OBJECTIVE SEARCH

Single-objective (accuracy) search is a special case of multi-objective search. For the proposed BOGCN-NAS, we randomly sample 50 architectures to fully train and use them as the initial trained architecture sets, which is counted into the total sample numbers. Since the whole NASBench and LSTM datasets can be inferred easily (less than 0.01s), we set $\mathcal{R} = \mathcal{A}$ in the experiment, which means we take total search space as the candidate pool. During the search phase, we use GCN to obtain embeddings and use the Bayesian regressor to compute *EI* scores for all architectures in the search domain, rank them based on the score and select the top ten models to fully train, obtain the accuracies and add them to trained architecture sets, update the best accuracy observed. The process is repeated for $k = 10$ times. The GCN predictor is then retrained with our updated trained architecture sets. This is repeated until the target model is found over 50 rounds. Note that GCN model is trained using Exponential Weighted Loss instead in NAS procedure.

BOGCN-NAS is compared with the following state-of-the-art sample-based NAS baselines: (i) Random Search, which explores the search space without performing exploitation; (ii) Regularized Evolution (Real et al., 2019b), which uses heuristic evolution process for exploitation but is still constrained by the available human prior knowledge; (iii) Monte Carlo tree search (MCTS) (Wang et al., 2019b); and (iv) LaNAS (Wang et al., 2019a). Both MCTS and LaNAS only estimate the performance of models in a coarse subspace, and then select models randomly in that subspace. In contrast, Bayesian optimization conducts a more fine-grained search and predicts the expected improvement of candidate pool.

Figure 3 and Table 2 show the number of samples until finding the global optimal architecture using different methods. The proposed algorithm outperforms the other baselines consistently on the two different datasets. On NASBench, BOGCN-NAS is $128.4\times$, $59.6\times$, $50.5\times$, $7.8\times$ more sample-efficient than Random Search, Regularized Evolution, MCTS and LaNAS respectively. On the smaller NLP dataset, BOGCN-NAS can still search and find the optimal architecture with fewer samples.

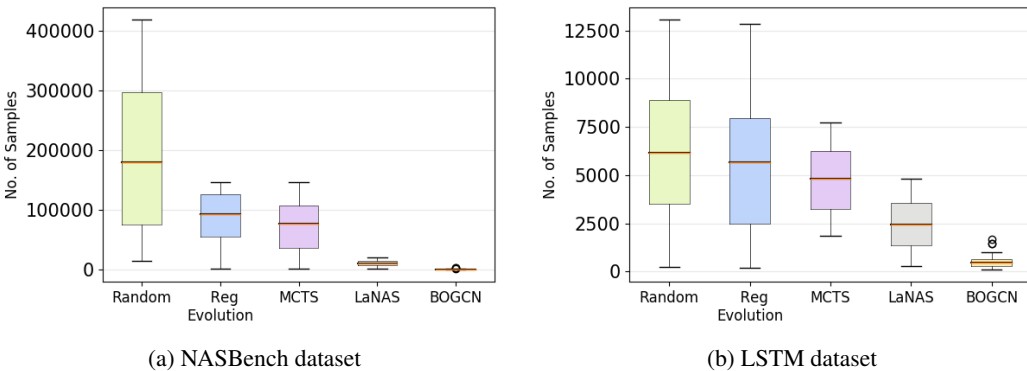

(a) NASBench dataset            (b) LSTM dataset

Figure 3: Comparison of number of architecture samples truly evaluated to find the globally optimal architecture over 50 rounds. On NASBench, BOGCN-NAS is $128.4\times$, $59.6\times$, $50.5\times$, $7.8\times$ more sample-efficient than Random Search, Regularized Evolution, MCTS and LaNAS.

We predict all architectures ($420K$; $12K$) together on NASBench and LSTM dataset because the cost of inference all once is negligible. For larger search space, we can use sampling methods as mentioned in Section 3.4 instead. In every iteration, we randomly sample a subset architectures $\mathcal{R}$ as candidate pool from the search space $\mathcal{A}$ for performance prediction and select top models among this pool. The performance of BOGCN versus the pool sampling ratio ($|\mathcal{R}|/|\mathcal{A}|$) is shown in Table 3. As can be seen, the overall performance of our BOGCN can still find the optimal model with less samples compared with other baselines. Even though random sampling is good enough, an alternative sampling method can be Evolutionary Algorithm.

Table 3: The performance with different pool sampling ratio.

| Sampling ratio | NASBench dataset | LSTM dataset |
|---|---|---|
| 1 | 1465.4 | 558.8 |
| 0.1 | 1564.6 | 1483.2 |
| 0.01 | 2078.8 | 1952.4 |
| 0.001 | 4004.4 | 2984.0 |

### 4.3 MULTI-OBJECTIVE SEARCH

In this section, we focus on multi-objective (accuracy and number of parameters) search. As the same settings of BOGCN-NAS with Section 4.2, the only difference is the criteria for updating the best architecture observed. Here we extend baselines in Section 4.2 to multi-objective form for comparison. In detail, we sample $2,000$ architectures in total and compare the found Pareto front $\widetilde{\mathcal{P}_f}$ with the optimal Pareto front $\mathcal{P}_f$ on NASBench dataset.

As shown in Figure 4a, the grey dots are the overall architectures of NASBench, the red dots are samples selected by BOGCN and the blue dots are architectures undominated by our selected samples. Based on them, we can make sure the respective Pareto fronts - the green dashed line is the optimal Pareto front and the red dashed line is the estimated Pareto front. Figure 4b shows the Pareto fronts estimated by different algorithms, which demonstrates the superiority of BOGCN mothod. Compared with other baselines, the models sampled by our method are gathered near the optimal Pareto front, and the found Pareto front is also closer to the optimal Pareto front. This validates the efficiency of our algorithm on this multi-objective search task.

### 4.4 OPEN DOMAIN SEARCH

In this section, we validate the propose method on the open domains - NASNet search space and ResNet style search space (described in Appendices Section). Since the size of open domain is

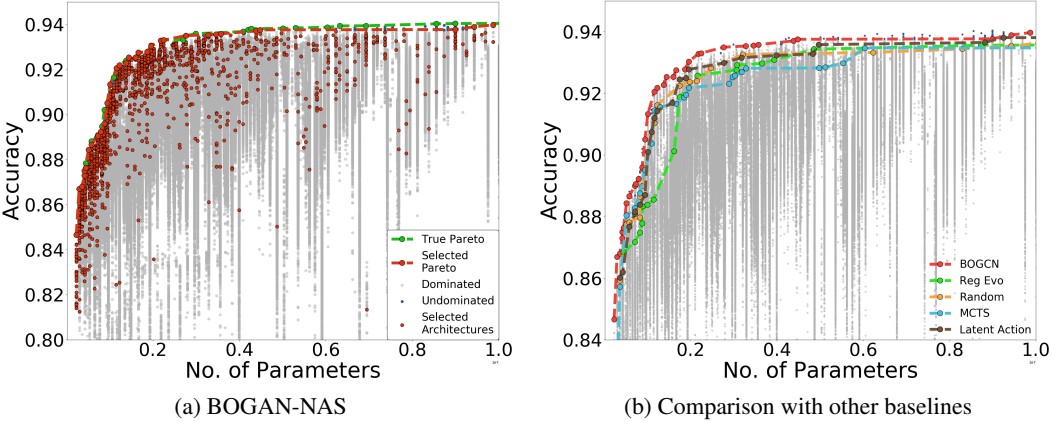

(a) BOGAN-NAS          (b) Comparison with other baselines

Figure 4: The result of multi-objective search with 2000 architecture samples. BOGAN-NAS can find more competitive Pareto optimal architectures.

enormous, we set the size of $\mathcal{R}$ equals to $1M$ every iteration for both search spaces. And the found architectures by BOGCN-NAS can be found in Appendices Section.

### 4.4.1 NASNET SEARCH SPACE

For NASNet search space, we consider single-objective (accuracy) search on Cifar-10. For efficiency, we train the sampled architectures by early stopping at 100 epochs rather than fully training. It's remarkable that we stop the algorithm after a certain number of samples $|S|$ rather than until finding the optimal architecture beacuse we don't know the optimal architecture and the open domain contains billions of architectures. And other experiment settings are the same with Section 4.2 as well.

We pick two best performing architectures V1 and V2 within 200 and 400 samples respectively and fully train them. Table 2 compares our results with other baselines on CIFAR-10. As can be seen, even though one-shot NAS methods don't need any architecture evaluated directly, the performance of the final found models is not as good as sample-based methods averagely. Compared with other sample-based NAS method, BOGCN outperforms all methods except for AmoebaNet-B, which costs $67.5\times$ more evaluating samples.

Table 4: Comparison of different methods on CIFAR-10.

| Model | Params | Top-1 err | No. of samples truly evaluated |
| --- | --- | --- | --- |
| NASNet-A+cutout (Zoph et al., 2018) | 3.3 M | 2.65 | 20000 |
| AmoebaNet-B+cutout (Real et al., 2019b) | 2.8 M | 2.55 | 27000 |
| PNASNet-5 (Liu et al., 2018a) | 3.2 M | 3.41 | 1160 |
| NAO (Luo et al., 2018) | 10.6 M | 3.18 | 1000 |
| ENAS+cutout (Pham et al., 2018) | 4.6 M | 2.89 | - |
| DARTS+cutout (Liu et al., 2019b) | 3.3 M | 2.76 | - |
| BayesNAS+cutout (Zhou et al., 2019) | 3.4 M | 2.81 | - |
| ASNG-NAS+cutout (Akimoto et al., 2019) | 3.9 M | 2.83 | - |
| BOGCN+cutout (V1) | 3.08M | 2.74 | 200 |
| BOGCN+cutout (V2) | 3.48M | 2.61 | 400 |

### 4.4.2 RESNET STYLE SEARCH SPACE

For ResNet style search space, we validate the proposed method for multi-objective search on ImageNet (Deng et al., 2009). Here we consider classification accuracy and the number of parameters of the model at the same time. Due to the large volume of the dataset, we train the sampled architectures by early stopping at $40$ epochs rather than fully training. And other experiment settings are the same with Section 4.3 as well.

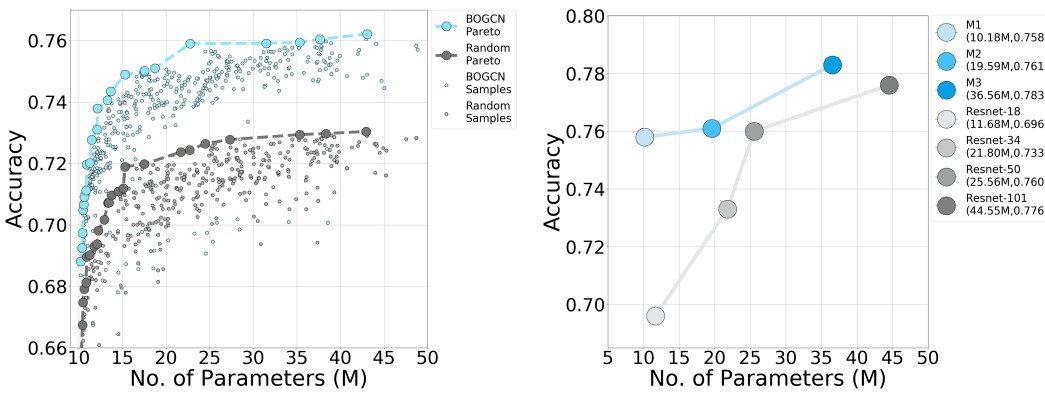

(a) Comparison of BOGCN-NAS and Random Search.   (b) Comparison of the found models and ResNets.

Figure 5: Open domain multi-objective search on ResNet style search space with 500 samples. Compared with Random Sampling, BOGCN-NAS achieves a more competitive Pareto front.

The accuracy and number of parameters of the sampled model are illustrated in Figure 5. Compared to random sampling, BOGCN-NAS achieves a more competitive Pareto front. We fully train every model on the estimated Pareto front and pick three models (M1, M2, M3), which can dominate ResNets. The comparison of our found models and famous ResNet family models are shown in Figure 5b. It shows that ResNets can be dominated by our found models seriously.

### 4.5   SEARCH SPCAE EXTENSION TRANSFER

In this section, we discover the ability of our proposed algorithm on search spcae extension transfer. Most NAS algorithms only focus on static search space. In contrast, how to adapt the methods for extension of search space is still an open problem. For instance, after searching in one small search space $\mathcal{A}_1$, how to transfer the obtained knowledge to a larger search space $\mathcal{A}_2$. For validation, we split NASBench into two sub-datasets: architectures with 6 nodes ($62K$) and architectures with 7 nodes ($359K$).

Using the same settings with Section 4.2, we pretrained our GCN model on architectures with 6 nodes and then transfer the GCN predictor to searching on the architecture domain with 7 nodes. For comparison, we run the same algorithm without pretraining. The search method with pretrained GCN predictor finds optimal model after 511.9 samples while the method without pretraining needs 1386.4 samples. As can be seen, pretraining can reach the optimal model $2.7\times$ more efficiently than unpretrained, validating the transfer ability of GCN predictor. Thus, BOGCN-NAS can handle different-scale architectures as long as their operation choices are the same.

## 5   CONCLUSION

In this work, we propose BOGCN-NAS, a multi-objective NAS method using Bayesian Optimization with Graph Convolutional Network predictor. We formulate the problem as a multi-objective optimization problem and utilize the efficiency of BO to search top-performance architectures. Instead of using popular Gaussian Processes surrogate model, we replace it with proposed GCN predictor such that graph-based structure can be preserved better. For NAS-specific tasks, we also propose weighted loss focusing on top-performance models. Experimental results show that the proposed algorithm outperforms the SOTA LaNAS on single-objective NAS and validate its efficiency on multi-objective NAS as well.

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

# Appendices

## A ABLATION STUDY

### A.1 THE ADVANTAGE OF GCN PREDICTOR AND BAYESIAN OPTIMIZATION

To verify that GCN is a superior choice of predictor against others (e.g. MLP, RNN), we replaced GCN in the search algorithm with MLP predictor. And for demonstrating that Bayesian Optimization indeed improves the performance of our search algorithm, we also remove BLR and use point estimation only. In other words, we select candidate models based only on GCN's predictive accuracies of architectures. Therefore, there are four different models correspondingly: (1) MLP; (2) BOMLP; (3) GCN; (4) BOGCN, where MLP, GCN use predictor only for model selection and BOMLP, BOGCN apply Bayesian Optimization on the strength of respective predictor.

With other settings same with with Section 4.2, we perform single-objective NAS algorithms over 50 rounds and the results of experiments on NASBench and LSTM datasets are shown in Figure 6. As can be seen, GCN is able to discover the optimal architecture with fewer samples than MLP, which proves the superiority of GCN predictor and our algorithm is more efficient with Bayesian Optimization.

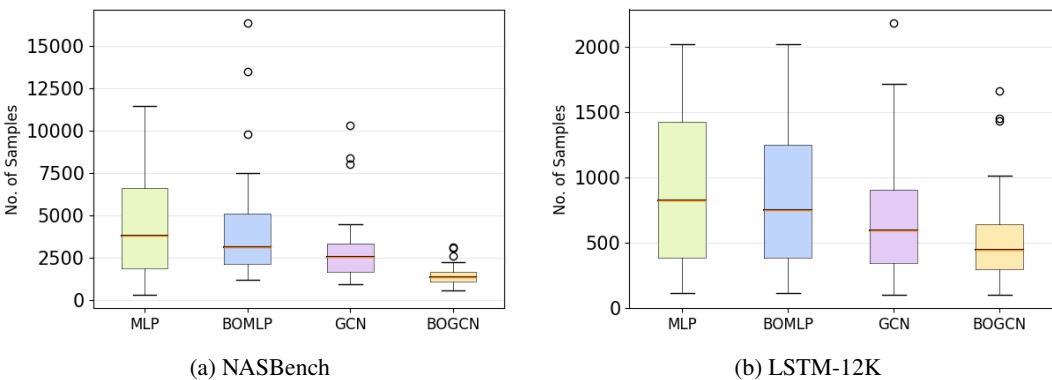

(a) NASBench          (b) LSTM-12K

Figure 6: Ablation study of Bayesian Optimization and Graph Convolutional Network Predictor. The comparison shows that the efficiency of the search can be boosted by adding Bayesian optimization and Graph Convolutional Network modules respectively.

### A.2 THE IMPROVEMENT OF EXPONENTIAL WEIGHTED LOSS

To prove the improvement of our proposed weighted loss empirically, we compare it with MSE loss. We also design other two weighted losses as following to show the influence of the second-order derivative of added weight. We apply the same settings with Section 4.2 for algorithm performing and the experiment result of single-objective (accuracy) on NASBench is shown in Figure 7. As can be seen, exponential weighted loss outperforms other three losses, which is consistent with our intuition.

$$L_{log} = \frac{1}{N \log 2} \sum_{i=1}^{N} \log(\widetilde{y_i} + 1)||y_i - \widetilde{y_i}||^2, \tag{10}$$

$$L_{linear} = \frac{1}{N} \sum_{i=1}^{N} \widetilde{y_i}||y_i - \widetilde{y_i}||^2. \tag{11}$$

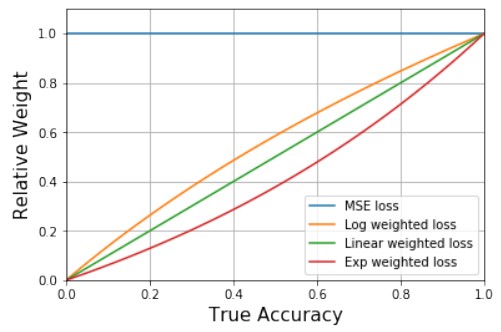 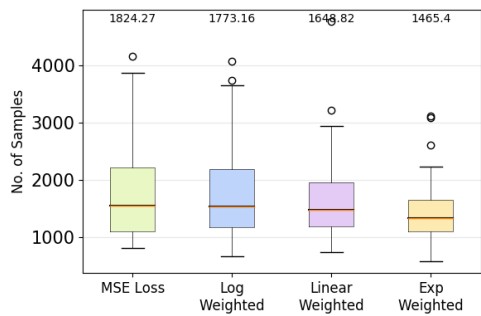

(a) The illustration of four losses considered in this paper.

(b) The performance of four losses over 50 rounds. It can be found that our method can reach the highest efficiency with exponential weighted loss.

Figure 7: Ablation study of Weighted Loss.

# B    SUPPLEMENTAL EXPERIMENTS

## B.1    TIME COURSE PERFORMANCE COMPARISON

Following (Wang et al., 2019b;a), besides comparing the number of training architectures sampled until finding the global optimal architecture, we also evaluate the current best models during the searching process. As shown in Figure 8, our proposed BOGCN-NAS outperform other search algorithms except for the very beginning on LSTM model searching.

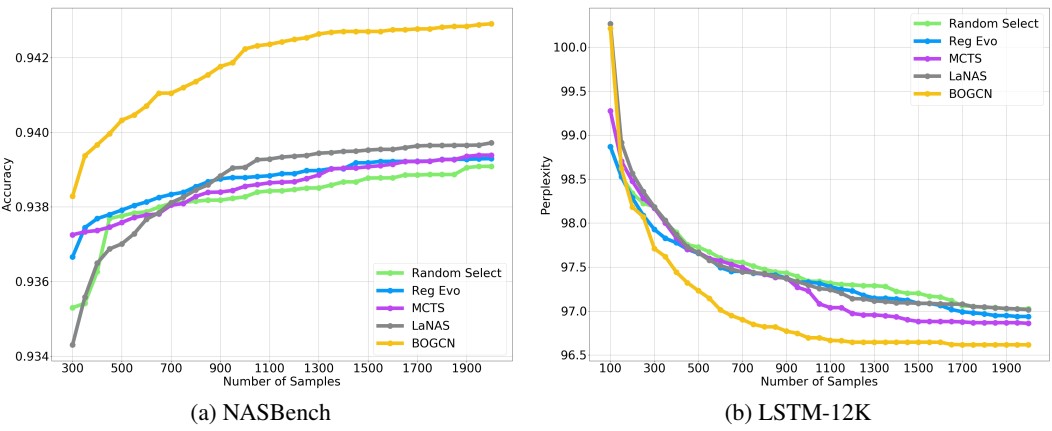

(a) NASBench

(b) LSTM-12K

Figure 8: Current best accuracy during searching. The result demonstrates that our proposed BOGCN-NAS outperforms other search algorithms significantly.

## B.2    PREDICTOR TRAINING ON WHOLE SEARCH SPACE

For comparison with previous work (Wang et al., 2019b), we also train our GCN predictor on whole NASBench dataset ($420K$ models) (Ying et al., 2019). We use $85\%$ NASBench for training, $10\%$ for validation and remaining $5\%$ for testing. As shown in Figure 9 (the value also means the density of architectures around), GCN outperforms others consistently with experiment training on fewer data (Section 4.1). Even though the correlations of the GCN and MLP are comparable here, the performance is less important than cases training on fewer data.

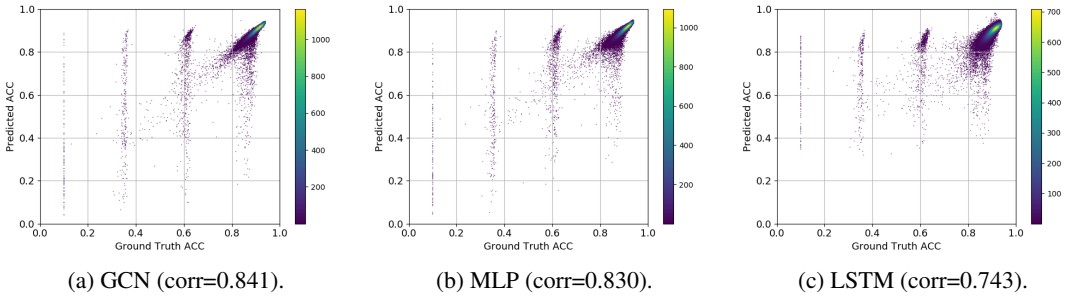

(a) GCN (corr=0.841).       (b) MLP (corr=0.830).       (c) LSTM (corr=0.743).

Figure 9: Correlation of predicted accuaracy versus ground truth for various predictors on whole dataset. It shows that GCN predictor can predict more accurately than other two predictors on whole NASBench.

## C DATASET AND SEARCH SPACE

### C.1 NASBENCH ENCODING

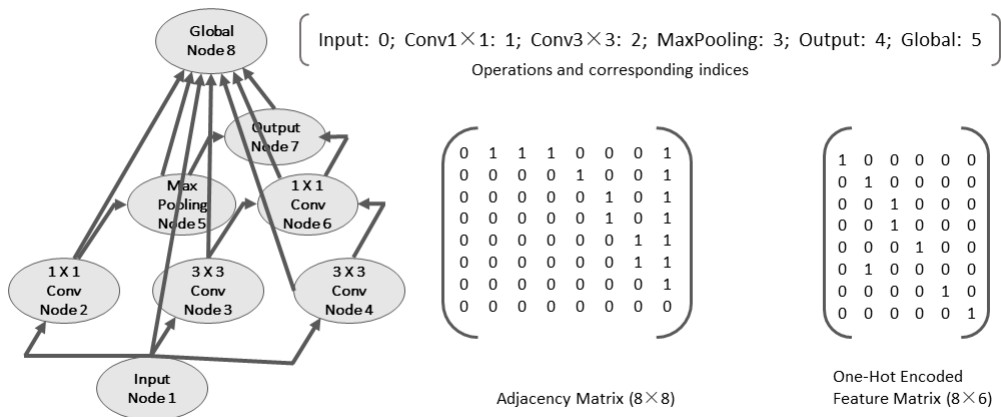

Figure 10: The encoding scheme of an example cell in NASBench.

### C.2 LSTM-12K DATASET

To create LSTM model dataset, we follow the same setting proposed by ENAS (Pham et al., 2018), we use adjacency matrix and a list of operators to represent an LSTM cell, and randomly sampled 12K cell structures from that domain, because the cells have natural graph structures, it is easy to feed them directly into GCN and conduct training. Due to the limitation on computational resources, we only sample architectures with number of nodes less than or equal to 8, and trained each cell for 10 epochs on PTB dataset (MARCUS et al., 1993). We use perplexity as a performance measure for the cells.

### C.3 NASNET SEARCH SPACE

We follow the search space setting of DARTS (Liu et al., 2019b), in which the architecture is stacked by the learned cell. The cell consists of 4 blocks, two inputs (the output of previous cell and the previous previous cell) and one output. There are 8 types operations allowed: $3 \times 3$ and $5 \times 5$ separable convolutions, $3 \times 3$ and $5 \times 5$ dilated separable convolutions, $3 \times 3$ max pooling, $3 \times 3$ average pooling, identity, and zero. Similar to previous work (Liu et al., 2018a), we apply the same

cell architecture for both "normal" and "reduction" layer. For adapting to proposed GCN predictor, we regard the operation as the node and regard the data flow as the edge. The encoding examples are illustrated in Appendices E.

### C.4 RESNET STYLE SEARCH SPACE

We follow the setting of Li et al. (2019) to prepare the ResNet style search space. This search space aims to find when to perform down-sampling and when to double the channels. The ResNet style backbone consists of 5 stages with different resolutions from input images. The spatial size of Stage 1 to 5 is gradually down-sampled by the factor of 2. As suggested in Li et al. (2019), we fixed one $3 \times 3$ convolution layer ($stride = 2$) in Stage-1 and the beginning of Stage-2. We use the block setting as bottleneck residual block in ResNet. Then, the backbone architecture encoding string looks like "$1211 - 211 - 1111 - 12111$", where "$-$" separates each stage with different resolution, "1" means regular block with no change of channels and "2" indicated the number of base channels is doubled in this block. The base channel size is $64$. In Section 4.4, we just take "$-, 1, 2$" as three different operations and encode architectures as a series of strings. We train the model generated from this search space for $40$ epochs with a fast convergence learning rate schedule. Each architecture can be evaluated in $4$ hours on one server with $8$ V100 GPU machines.

## D NAS RELATED WORK

NAS aims at automatically finding a neural network architecture for a certain task such as CV and NPL (Chen et al., 2018; Liu et al., 2019a; Chen et al., 2019) and different datasets without human's labor of designing networks. Particularly, in real applications, the objective of NAS is more preferred to be obtaining a decent accuracy under a limited computational budget. Thus a multi-objective NAS is a more practical setting than only focusing on accuracy. There are several approaches in NAS area: 1) Reinforcement learning-based algorithm. Baker et al. (2016); Zoph & Le (2017); Cai et al. (2018) train an RNN policy controller to generate a sequence of actions to design cell structure for a specify CNN architecture; 2) Evolution-based algorithm. Real et al. (2017); Liu et al. (2018b); Real et al. (2019a) try to evolve architectures or Network Morphism by mutating the current best architectures and explore new potential models; 3) Gradient-based algorithm Liu et al. (2019b); Cai et al. (2019); Xie et al. (2019) define an architecture parameter for continuous relaxation of the discrete search space, thus allowing differentiable optimization of the architecture. 4) Bayesian Optimization-based algorithm. (Kandasamy et al., 2018) and (Jin et al., 2019) define the heuristic distances between architectures and apply BO with Gaussian Processes. Among those algorithms, most existing methods focus on a single objective (accuracy), others adding computation constraints as a regularization loss in the gradient-based method or as a reward in the RL-based algorithm. In contrast, our method reformulates the multi-objective NAS as a non-dominate sorting problem and further enables an efficient search over flexible customized search space.

## E BEST FOUND MODELS

### E.1 NASNET SEARCH SPACE

Figure 11a and 11c show the found architectures from the open domain search on NASNet search space by our method.

### E.2 RESNET STYLE SEARCH SPACE

Figure 12, 13 and 14 show the found architectures from the open domain search on ResNet style search space by our method.

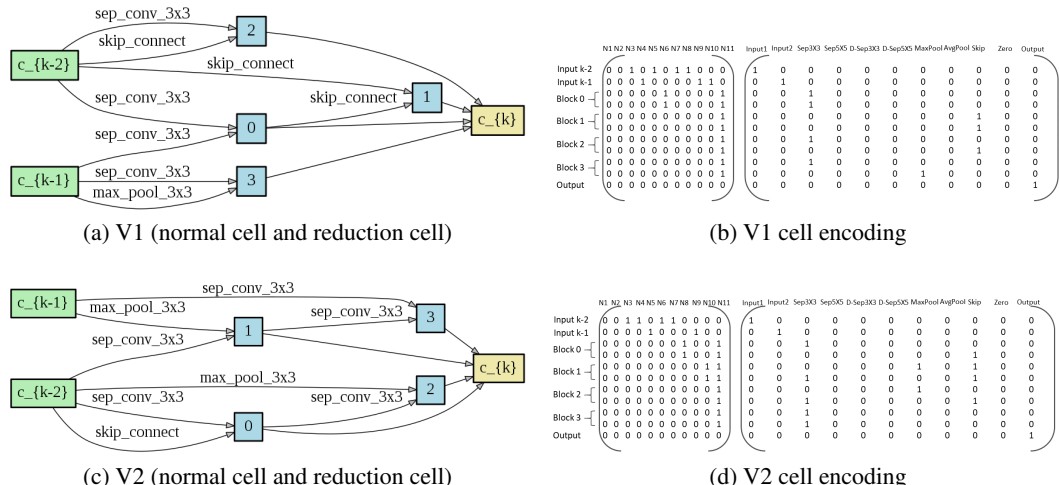

(a) V1 (normal cell and reduction cell)

(b) V1 cell encoding

(c) V2 (normal cell and reduction cell)

(d) V2 cell encoding

Figure 11: The found models in NASNet search space.

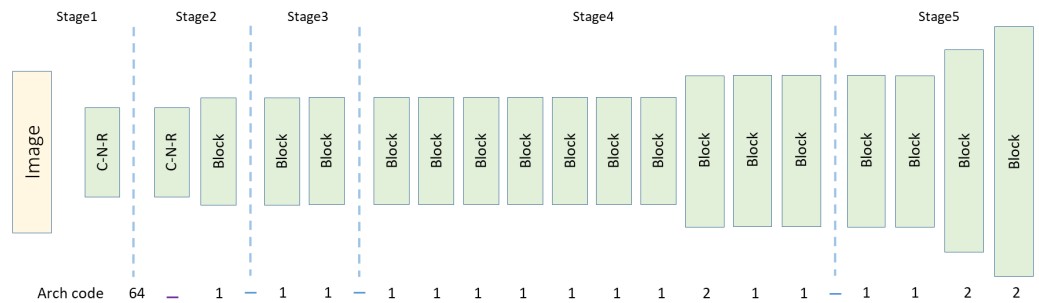

Figure 12: M1 ("1 − 11 − 1111111211 − 1122")

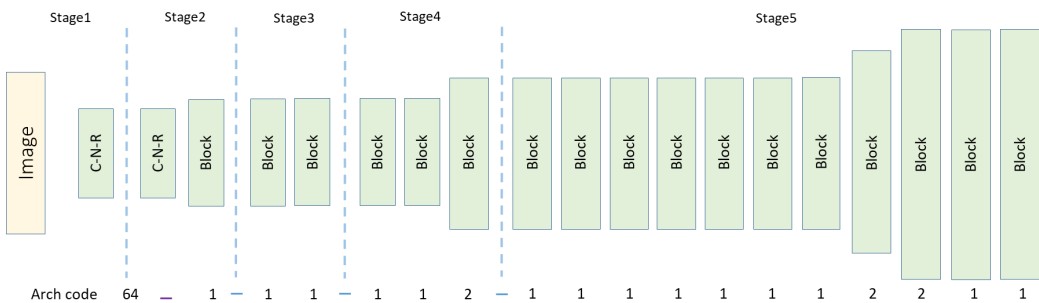

Figure 13: M2 ("1 − 11 − 112 − 11111112211")

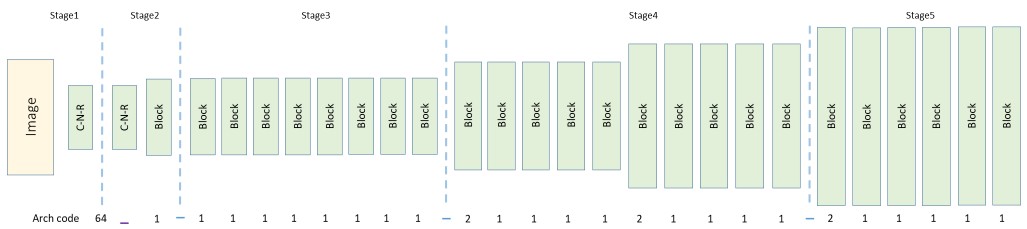

Figure 14: M3 ("1 − 11111111 − 2111121111 − 211111")

