# OpenReview forum: "Multi-objective Neural Architecture Search via Predictive Network Performance Optimization"
_ICLR.cc/2020/Conference — Reject_

### Official Review · AnonReviewer2 · 2019-10-21
**Official Blind Review #2**

**Rating:** 3

**Review:**

This paper provide a NAS algorithm using Bayesian Optimization with Graph Convolutional Network predictor. The method apply GCN as a surrogate model to adaptively discover and incorporate nodes structure to approximate the performance of the architecture. The method further considers an efficient multi-objective search which can be flexibly injected into any sample-based NAS pipelines to efficiently find the best speed/accuracy trade-off.

The paper is well-written. The experiments are abundant. However, the paper has following drawbacks that need to be further concerned:

1.	In my opinion, the key point of the paper has nothing to do with GCN or multi-objective. The important part is to use BO and EI to sample new architecture. However, no theoretical proof is provided to guarantee that the performance is getting better during while loop in Algorithm 1.
2.	Eq.(9) focuses more on models with higher accuracy. However, those models with bad performance will be predicted inaccurately and may have a higher score than good models. For model with ground-truth near 0, arbitrary predicted score results in the same loss. Eq.(9) seems cannot prevent this situation from happening.
3.	Table 1 compare different architectures with GCN. However, the LSTM is the worst architecture used among the 3 different architectures in the original paper (Alpha-X), which makes the comparison unfair.
4.	Table 2 shows the number of architectures trained. However, the proposed method need to update GCN multiple times during searching, which makes the comparison unfair.
5.	Table 2 shows the number of training models before finding the best model. It is meaningless when used in reality, which often contains more than 10^10 different architectures and the best architecture is unknown. In my opinion, the performance of the top1 architecture predicted by the proposed method is much important.
6.	Algorithm 1 uses Pareto front, which does not exist when doing experiments on single-objective search. More details should be clarified.

**Experience Assessment:**

I have published one or two papers in this area.

**Review Assessment: Checking Correctness Of Derivations And Theory:**

I carefully checked the derivations and theory.

**Review Assessment: Checking Correctness Of Experiments:**

I carefully checked the experiments.

**Review Assessment: Thoroughness In Paper Reading:**

I read the paper thoroughly.

---

> ### Author Response · Authors · 2019-11-15
> **Reply to review #2**
>
> Thank you for your comments. We address your main concerns as follows.
>
> 1. "the key point of the paper has nothing to do with GCN"
> - We believe GCN predictor is one of the key points of our paper. As we mentioned, GCN predictor is proposed as the surrogate model for Bayesian Optimization. Different from popular Gaussian Processes, GCN predictor can obtain the architecture embeddings better as shown in the experiments in Figure 2.
>
> 2. "the key point of the paper has nothing to do with multi-objective"
> - Multi-objective search is not discussed in some well-known NAS papers such as DARTS, ENAS, NAO, etc. We believe it is not fully explored in the NAS literatures. We have shown that our method is much better than other multi-objective searching algorithms by adding more baseline comparisons in Section 4.3.
> https://drive.google.com/file/d/1xs4SNva5hdd7Rhaok15cPP7UXohVi5QR/view?usp=sharing
>
> 3. "no theoretical proof is provided to guarantee that the performance is getting better during while loop in Algorithm"
> - Bayesian Optimization and Graph Convolutional Network are both well explored theoretically. For BO part, it has theory for balancing exploitation and exploration [1]; For GCN part, it has theory for guaranteeing the better embeddings for graph-based data [2].
>
> 4. "those models with bad performance will be predicted inaccurately and may have a higher score than good models."
> - We find that bad models will still have a lower score than good models. For the NAS problem, we only care about those high-accuracy models. Our motivation is to distinguish low-performance model roughly. For example, the model with lowest accuracy in NASBench (true accuracy=0.0998) will be predicted to estimated accuracy=0.2872, which is still separable from top models (estimated accuracy=0.94). Adding more weights on models with high performance doesn't mean we don't care models with bad performance.
>
> 5. "For model with ground-truth near 0, arbitrary predicted score results in the same loss."
> - Even though the loss w.r.t. $\widetilde{y_i}=0$ won't be counted, it won't be predicted so high. Furthermore, the ablation study section is shown in Appendices section (A.2). Empirically, adding the weighted loss will speed up 1.25X for searching the optimal on NASBench.
>
> 6. "the LSTM is the worst architecture used among the 3 different architectures in the original paper (Alpha-X), which makes the comparison unfair."
> - We did not only compare GCN with LSTM, MLP (the best predictor in Alpha-X) is also included in the comparison. Furthermore, we used MLP for all following ablations studies (Appendices A.1). There are two different architectures in Alpha-X because multi-stage is the ensemble form, which can also be used in GCN.
>
> 7. "the proposed method needs to update GCN multiple times during searching, which makes the comparison unfair."
> - For NAS problem, the cost bottleneck is the evaluation of the architecture. Compared with evaluation cost, GCN updating cost is negligible. Other baselines have similar steps. For instance, previous SOTA LaNAS [3] need multiple times for action space updating.
>
> 8. "It is meaningless when used in reality, which often contains more than 10^10 different architectures and the best architecture is unknown"
> - It is one common metric for NAS problem. Previous SOTA LaNAS [3] and MCTS [4] both compare this metric.
>
> 9."the performance of the top1 architecture predicted by the proposed method is much important."
> - We indeed compared time course performance of different methods (please see appendices B.1). Our method outperforms other baselines consistently.
> https://drive.google.com/file/d/1IOr511FIjCIfIxeqLn1JIZmV0FKuog-K/view?usp=sharing
>
> 10. "Algorithm 1 uses Pareto front, which does not exist when doing experiments on single-objective search."
> - Single-objective problem is compatible with our definition of multi-objective problem (Section 2.1) where the Pareto front is only constituted by one architecture. We describe single-objective case specifically in the updated version.
>
> [1] A tutorial on Bayesian optimization of expensive cost functions, with application to active user modeling and hierarchical reinforcement learning. 2010
> [2] Semi-Supervised Classification with Graph Convolutional Networks. 2017
> [3] Sample-Efficient Neural Architecture Search by Learning Action Space. 2019
> [4] AlphaX: eXploring Neural Architectures with Deep Neural Networks and Monte Carlo Tree Search. 2019

---

### Official Review · AnonReviewer3 · 2019-10-23
**Official Blind Review #3**

**Rating:** 3

**Review:**

This paper proposed BOGCN-NAS that encodes current architecture with Graph convolutional network (GCN) and uses the feature extracted from GCN as the input to perform a Bayesian regression (predicting bias and variance, See Eqn. 5-6). They use Bayesian Optimization to pick the most promising next model with Expected Improvement, train it and take its resulting accuracy/latency as an additional training sample, and repeat.

They tested the framework on both single-objective and multi-objective tasks.  On the single-objective (accuracy task). They tested it on NasBench and LSTM-12K, two NAS datasets with pre-trained models and their performance. They obtained very good performance on both, beating LaNAS (previous SoTA) by 7.8x higher sample efficiency. On multiple-objective, they show higher efficiency in finding Pareto frontier models, compared to random search.

One main question I have is whether the next model is chosen given the current prediction model? For NasBench, did you run your predictor for all (420k minus explored) models and pick the one that maximizes Expected Improvement? Note that LaNAS is more efficient in that manner by sampling directly on polytopes formed by linear constraints. If so, how do you pick the next candidate models in open domain setting?

It looks like Eqn. 9 biases the training heavily towards high accuracy region, which is a hack. Although in the Appendix (Fig. 7) the authors have already perform some analysis on the effect of different weight terms, I wonder whether there is a more problem-independent way of doing it. The MCTS in LaNAS is one way that automatically focuses the search on the important regions. Currently, the proposed approach might limit the usability of the proposed method to other situations when accuracy is no longer that important.

The performance is really impressive in the NasBench and LSTM dataset. The paper mentioned that “BO can really predict the precious model to explore next” but didn’t provide an examples in the paper. I would whether the author could visualize it in the next revision, which would be very cool.

Do you have open domain search research in single-objective search?

Why not use NasNet architecture for a fair comparison with other NAS papers?

In Appendix, Fig. 6 shows that even without GCN and BO, a single MLP already achieves global optimal solution in LSTM-12K dataset with ~850 samples, already beating all the previous methods (Random, Reg Evolution, MCTS and LaNAS). If that’s the case, I wonder how much roles the proposed methods (BO + GCN) play during search? Also what do you mean by “without BO”? Do you only predict the mean and assume all variance is constant?

=====Post Rebuttal======
I have read other reviewers' comments and the rebuttal.

One of the main problems in this paper is an unfair comparison against LaNAS. LaNAS only uses a single sample at each leaf, while they sample 100% to 0.1% of the models, evaluate them with the current BO model and find the best. For NasBench-101 with 420K models, even sampling 0.1% each time means ~400 samples and the performance (from the rebuttal) seems to degrade substantially from 100% case (1464.4->4004.4). This means that almost 3x more samples are needed, compared to what they claimed.

I agree with the authors that calling BO function is super fast so maybe this is fine. However, on the open domain experiments, their performance is also not better than LaNAS+c/o, which they didn't list in the rebuttal. I listed it here:

Model                      	Params       	Top-1 err	No. of samples truly evaluated
-----------------------------------------------------------------------------------------------------------------
BOGCN+cutout (V1)	   3.1M 	          2.74	                      200
BOGCN+cutout (V2)	   3.5M 	          2.61	                      400
LaNAS+c/o                       3.2M              2.53±0.05                    803

Overall this paper is on the borderline. I don't mind if the paper gets rejected. For now I lower the score to 3.

**Experience Assessment:**

I have published one or two papers in this area.

**Review Assessment: Checking Correctness Of Derivations And Theory:**

N/A

**Review Assessment: Checking Correctness Of Experiments:**

I carefully checked the experiments.

**Review Assessment: Thoroughness In Paper Reading:**

I read the paper at least twice and used my best judgement in assessing the paper.

---

> ### Author Response · Authors · 2019-11-15
> **Reply to review #3**
>
> Thanks for your interests and we appreciate your constructive comments. We hope our following clarifications can address your concerns.
>
> 1. "whether the next model is chosen given the current prediction model, how do you pick the next candidate models in open domain setting?"
> - As mentioned in section 3.4, we randomly sample a subspace as candidate pool if the search space is huge. For NASBench, since the search space only contains 420K models, which can be inferred easily with very little cost (less than 0.01 seconds), we take the whole search space as our candidate pool. To simulate the situation in a very large search space, we also test our algorithm with different pool sampling ratio (1, 0.1, 0.01, 0.001) and the result is shown as follows. Even though we select a subspace for prediction, the improvement is still significant.
> ---------------------------------
> ratio  NASBench   LSTM
> 1           1465.4        558.8
> 0.1        1564.6      1483.2
> 0.01      2078.8      1952.4
> 0.001    4004.4      2984.0
> ----------------------------------
>
> 2. "It looks like Eqn. 9 biases the training heavily towards high accuracy region, which is a hack"
> - The main purpose of NAS is to search an architecture with high accuracy. Similar with MCTS in LaNAS, Our proposed loss is another way that focuses the search on the important regions. Even the weighted loss is removed, our method is still outperform than previous SOTA - LaNAS (shown in appendices section A.2).
>
> 3. "I wonder whether there is a more problem-independent way of doing it"
> - Our proposed weighted loss is a possible way to solve searching problem with attention, which is problem-independent. Our idea is inspired by Focal Loss [1], while Focal Loss is used for classification and our weighted loss is used for regression.
>
> 4. "Do you have open domain search research in single-objective search?"
> - Yes, the model we find can achieve 0.783 accuracy on ImageNet dataset. The model is the same as M3, but we can achieve it within less samples (400 samples) for single-objective search. We add experiment in DARTS search space, where each cell contains 4 blocks (11 nodes) and 8 possible operations. After sampling 200 models, we picked out 2 models with best test accuracy 97.26%, 97.39% on cifar10 respectively. The experiment details are appended in the updated paper
>
> 5. "Why not use NasNet architecture for a fair comparison with other NAS papers?"
> - We add experiment in DARTS search space, where each cell contains 4 blocks (11 nodes) and 8 possible operations. We picked out 2 models (V1 & V2) with best test accuracy on cifar10 after sampling 200 and 400 models respectively. The experiment details are appended in the updated paper.
> --------------------------------------------------------------------------------------------------------
> Model                      	Params	Top-1 err	No. of samples truly evaluated
> -------------------------------------------------------------------------------------------------------
> NASNet-A+cutout	  3.3 M	   2.65	                20000
> AmoebaNet-B+cutout	  2.8 M	   2.55	                27000
> PNASNet-5	                  3.2 M	   3.41	                1160
> NAO	                        10.6 M 	   3.18	                1000
> ENAS+cutout          	  4.6 M	   2.89	                    -
> DARTS+cutout	          3.3 M	   2.76	                    -
> BayesNAS+cutout	  3.4 M	   2.81	                    -
> ASNG-NAS+cutout	  3.9 M	   2.83	                    -
> --------------------------------------------------------------------------------------------------------
> BOGCN+cutout (V1)	  3.1M 	   2.74	                 200
> BOGCN+cutout (V2)	  3.5M 	   2.61	                 400
> ---------------------------------------------------------------------------------------------------------
>
> 6. "I wonder how much roles the proposed methods (BO + GCN) play during search?"
> - We indeed performed ablation study in appendices section (A.1 & A.2) and the number of samples evaluated is shown below. As can be seen, the improvement of BO and GCN is significant.
> ----------------------------------------------
>   MLP     BOMLP    GCN      BOGCN
> 4527.0   4042.25   2860.6    1465.4
> ----------------------------------------------
> On NasBench-101, compared to using only the GCN predictor, BOGCN finds global optimal with 1395.2 fewer samples; compared to using BOMLP, BOGCN finds global optimal with 2576.85 fewer samples, which proves the importance of BO and GCN respectively.
>
> 7. "Also what do you mean by “without BO”? Do you only predict the mean and assume all variance is constant?"
> - Yes, we remove BO part and select architectures only based on predictors. In this scenario, we use point estimation to predict the accuracy of each model, thus not taking variance into account. We make it clear in the updated version.
>
> [1] Focal Loss for Dense Object Detection. 2017

---

### Official Review · AnonReviewer1 · 2019-10-26
**Official Blind Review #1**

**Rating:** 3

**Review:**

--Summary--
The authors present an algorithm BOGCN-NAS which combines bayesian optimization and GCNs for searching over NN architectures. The authors emphasize that this method can be used for multi-objective optimization and run experiments over NAS-Bench, LSTM-12K and ResNet models.

--Method--

Methodologically, the contribution is somewhat weak. The main technical contribution is to use a GCN to get a global representation of a graph, which can then be used for downstream regression tasks such as predicting accuracy. It’s not clear how much the GCN generalizes in being able to encode arbitrarily large architectures. The two main examples offered are NAS-Bench and LSTM-12K focus on optimizing cell architectures which contain a handful of nodes e.g. (5 in the case of NAS-101).

Graph embeddings:
The authors have not considered other graph embeddings to use in their Bayesian regression setup. E.g. see https://arxiv.org/pdf/1903.11835.pdf for a list.

Bayes-opt:
In Algo1, step 5. The authors randomly sample a number of architectures in order to calculate EI scores on them. For large discrete combinatorial search spaces, this approach will not scale.

Multi-objective optimization:
It’s not clear why GCNs or BO is required for this. Any predictor that generates multiple metrics could substitute in order to create a pareto-curve. Even multi-objective RL based approaches could suffice. Thus multi-objective opt only seems like a minor/tangential contribution.

--Experiments--
The main claim of the paper is that this approach works well  for the multi-objective case. However, the results only look at two objectives #params vs accuracy. There’s a pretty strong correlation between the two. It’s unclear how the method generalizes when objectives are not correlated. The authors need to thoroughly demonstrate other objectives/find suitable benchmarks for the same as clearly NAS-101 will not suffice.

Other concerns:
 - Table 1 has correlations using 1000 training architectures. Why 1000? Why not 50 (that’s how sec 4.2 is initialized). Also, the correlation results are less impressive in Figure 9.
 - Table 1 lists the number of params that the predictor uses. Why is this important? How about comparing with a linear regressor?
 - The results in Sec 4.3 are using random as the only baseline. This is a pretty weak baseline.
 - In sec 4.4, the authors pick models M1, M2 and M3 as candidate examples.. How were these chosen ?
 - Sec 4.5 transfer learning results are pretty weak. Transfer across datasets is much more interesting e.g. between ImageNet and Cifar-10.

Overall, this paper has some interesting results, which show that GCNs can be useful models to encode graph structured inputs. However, the methodological and experimental results can definitely be strengthened. The authors may consider the following:
Address how GCNs can model and scale to general architecture spaces than a small number of nodes in a cell.
Address how to sample better over combinatorial search spaces than random in the inner loop of BO.
Strengthen MO-opt results. Use better baselines than random and different objectives than accuracy vs #params.


**Experience Assessment:**

I have published one or two papers in this area.

**Review Assessment: Checking Correctness Of Derivations And Theory:**

I assessed the sensibility of the derivations and theory.

**Review Assessment: Checking Correctness Of Experiments:**

I carefully checked the experiments.

**Review Assessment: Thoroughness In Paper Reading:**

I read the paper at least twice and used my best judgement in assessing the paper.

---

> ### Author Response · Authors · 2019-11-15
> **Reply to Review #1 [2/2]**
>
> Other Concerns:
> 5. "Table 1 has correlations using 1000 training architectures. Why 1000? Why not 50 (that's how sec 4.2 is initialized)."
> - In table 1, our purpose was to show that GCN's prediction accuracy is better comparing to other methods given few data.
>
> We have done experiments with 50, 550, 1050 ... 7050 training samples to simulate the actual search process. It can be found that our methods also works well in these situations.
>
> -------------------------------------------
> training archs GCN MLP LSTM
> 50                    0.385 0.082 0.209
> 550                  0.570 0.329 0.352
> 1050                0.597 0.414 0.472
> ...
> 7050                0.692 0.573 0.504
> --------------------------------------------
>
> 6. "Table 1 lists the number of params that the predictor uses. Why is this important? How about comparing with a linear regressor?"
>
> - We think the number of parameters of predictor is important, and predictors with fewer parameters are more efficient due to the following reasons:
>
> - Given fewer training data, predictors with more parameters tend to under-fit. In practice, we can only have very few trained models in the beginning.
>
> - The latency caused during prediction is shorter, which allows BOGCN to predict more models each time.
>
> - We have followed your advice and done the experiment with a linear regressor. The correlation is only 0.34, which is uncompetitive becasue it cannot handle non-linear features.
>
> -------------------------------------------
> GCN MLP LSTM Linear_Regressor
> 0.61  0.40   0.46          0.34
> --------------------------------------------
>
> 7. "The results in Sec 4.3 are using random as the only baseline. This is a pretty weak baseline."
> - As pointed out by [2, 3, 4], random search may not be a weak baseline in NAS problem. However, we agree with you about the lack of comparisons in multi-objective tasks. We have performed experiments using other methods and appended them in the paper.
>
> 8. "In sec 4.4, the authors pick models M1, M2 and M3 as candidate examples.. How were these chosen?"
> - We fully train every model on the estimated Pareto front and compare them with ResNets. Then we can pick three models (M1, M2, M3), which can dominate ResNets. We have elaborated more in detail in the paper.
>
> 9. "Sec 4.5 transfer learning results are pretty weak. Transfer across datasets is much more interesting e.g. between ImageNet and Cifar-10."
> - The motivation of this experiment is to provide evidence for expanding search space dynamically. We want to show that GCN can adapt to architecture cells with a varying number of nodes. Using this feature, we could progressively search architectures: start with a small number of nodes and gradually grow to larger architectures. We will consider doing transfer learning across different datasets, but this kind of experiment is very time consuming, thus can only be further explored in future works.
>
> [1] Semi-Supervised Classification with Graph Convolutional Networks. 2017
> [2] Evaluating The Search Phase of Neural Architecture Search. 2019
> [3] Random Search and Reproducibility for Neural Architecture Search. 2019
> [4] Exploring Randomly Wired Neural Networks for Image Recognition. 2019

---

> ### Author Response · Authors · 2019-11-15
> **Reply to Review #1 [1/2]**
>
> Thank you for spending time to read our paper, we find your comments very insightful and constructive. We improved our paper according to your reviews.
>
> 1. "It's not clear how much the GCN generalizes in being able to encode arbitrarily large architectures"
> - GCN can handle a large number of nodes. For instance, the Pubmed citation dataset contains 19,717 nodes and GCN can encode it pretty well [1]. One of the most notable features of GCN is its ability to encode graphs with varying sizes, as long as the possible operations are the same. We also added experiment in DARTS search space, where each cell contains 11 nodes and 8 possible operations. After sampling 200 models, we picked out 2 models with best test accuracy 97.26%, 97.39% respectively. The experiment details are appended in the updated paper.
>
> 2. "For large discrete combinatorial search spaces, this approach will not scale."
> - During the open-domain experiment on DARTS search space, which contains over $10^{12}$ architectures, we re-sample 1M architectures randomly after fully training every 5 models, we found that the time used to predict the architectures is negligible(less than 0.01 seconds). We also performed experiments on NASBench-101, sampling 10%, 1% and 0.1% of the search space respectively, the results are still much better than other methods. It shows that the predictive power of BOGCN is quite reliable, it can pick out good candidates as well as exploring new ones with only a few fully-trained models.
>
> 3. “multi-objective opt only seems like a minor/tangential contribution."
> - Multi-objective search is not mentioned in some well-known NAS paper like DARTS, ENAS, NAO, etc. We believed it is not fully explored in the NAS literatures. We have shown that our method is better than other multi-objective searching algorthim by adding more comparision in Section 4.3. For visualization: https://drive.google.com/file/d/1xs4SNva5hdd7Rhaok15cPP7UXohVi5QR/view?usp=sharing
>
> 4. “There’s a pretty strong correlation between the parameter and accuracy”
> - We believe model accuracy and number of parameters don't necessarily have a strong correlation. It can be seen in Figure 4, many models share the same number of parameters while their accuracies are quite different. The correlation between accuracy and No.of parameters in Figure 4 is only 0.145, which is a pretty weak correlation.

---

### Public Comment · ~Linnan_Wang1 · 2019-10-01
**Interesting work**

The results are truly impressive, but I have a question, it will be great if the authors can clarify.

In Fig.2, it looks that LSTM actually is the best prediction model since most points are aligned with y = x. However, in your table 1. GCN (corr=0.819) greatly improved w.r.t MLP(0.522) and LSTM (0.4). Could you please explain why?

---

> ### Author Response · Authors · 2019-10-03
> **Table 1 is consistent with Figure 2**
>
> Thanks for your comments.
>
> Table 1 is indeed consistent with Figure 2. For the LSTM plot, most points (lightest part) are below the line "y=x", while for the GCN plot, most points are close to this line. For better illustration, we will add the line "y=x" to Figure 2 in the final version.

---

### Public Comment · ~Scarlett_Li1 · 2019-10-10
**Question about performance of predictors**

I have done some similar experiments to build a performance predictor. But I found if we using some basic feature of models(like the number of Conv, the number of operations), and use the GBDT to predict the performance of the model, it could beat the models that based on Neural Network(whatever GCN/LSTM/MLP) in limited data.

So could you do the simple experiments to prove the performance of your model?  Could you share with us?

And have you tried to compare your method with the STOA NAS methods(like ENAS/DARTS/ONE-SHOT/PROXYLESSNAS)?
Because random search is not a strong baseline.
Could you share with us?

Another question is:
As we all know, the more data(arch) you have the better performance of preditor.
But the practical NAS is a search progress, which means the predictor may collect many the training samples with low accuracy but fewer samples with high accuracy(or other metrics).   Which means that the predictor may perform poorly on high accuracy model predictions(But this is the part we really care about).
As this paper mentioned, "We used 1000 architectures
in NASBench for training, 100 architectures for validation, and 10000 architectures for testing". I wonder how to select the test sample?

Thank you.

---

> ### Author Response · Authors · 2019-10-23
> **Our GCN predictor performs better than GBDT predictor**
>
> Thanks for your comments.
>
> - Following your idea using GBDT predictor with basic features for NAS, we have performed the experiment on NASbench [1] and compared it with using only GCN predictor (described in Appendices A.2). However, the result of GBDT predictor is poor. The model struggles to find the optimal architecture given 20000 samples while GCN can find the optimal just given 2500 samples.
> - Intuitively, only given the number of each operation is not sufficient to uniquely determine a model, which causes many models to have the same predicted accuracy. Furthermore, the connections between those operations are also essential. The reason why GBDT is superior in your experiment may be that the search space is too small, and there aren’t many architectures with the same number of each operation.
>
> - We have compared our proposed model with SOTA algorithm [2] described in section 4.2. Note that in the introduction section, we make it clear that one-shot models (like ENAS/DARTS/ONE-SHOT/PROXYLESSNAS) cannot find the optimal models because of weight sharing or  continuous relaxation. And the motivation of our method is to find the optimal model with less cost. Also, following the evaluation method in previous papers [2, 3], it excludes one-shot methods because they just find the competitive model rather than optimal model.
>
> - We totally agreed with your concern about high accuracy model. Therefore, we proposed Exponential Weighted Loss in section 3.5, which focuses more on high accuracy model (the ablation study is shown in Appendices A.3).
> - We have described the practical NAS is an online learning progress in section 3.4. Thus in experiment, we started with 50 initialized data (available easily) and the final result is good enough. As you said, the predictor is not good initially, but it can predict quite well after a small number of samples.
> - For section 4.1, we select three datasets (train/validation/test) randomly. Here we just want to prove the performance of predictors with few training data (1K) rather than total dataset (420K).
>
> [1] NAS-Bench-101: Towards Reproducible Neural Architecture Search. ICML 2019
> [2] Sample-Efficient Neural Architecture Search by Learning Action Space. arXiv preprint
> [3] AlphaX: eXploring Neural Architectures with Deep Neural Networks and Monte Carlo Tree Search. arXiv preprint

---

### Public Comment · ~Linnan_Wang1 · 2019-10-23
**Fail to replicate the Fig.6**

Hello there,

I have implemented a MLP on NASBench, and the source code is available at: https://github.com/linnanwang/MLP-NASBench-101

After running the code 100 times, I get an average number of 50000 samples to get the global optimum. This is far larger than your result ( 3000 ~ 4000 avg samples to the global optimum in Fig.6(a), MLP). It is possible that the discrepancy results from the hyper-parameters in my codes. If you find something can significantly improve my code, please advise.

Otherwise, I suspect your results are not from different random seeds. Picking a good random seed that gives fewer samples to the global optimum makes the entire results unfair for comparison. Please clarify. Thank you.

---

> ### Author Response · Authors · 2019-10-23
> **You didn’t add activation functions in your MLP and didn't set gradients to zero every iteration**
>
> We checked your code and made the following changes to your code:
> 1. We added Relu activation after the first layer and sigmoid after the second layer.
> 2. We have set optimizer gradient to 0 in every iteration during training.
> 3. We train the mlp predictor from scratch every time after sampling new data, number of epochs is set to be 150 and lr is 0.001.
>
> After making the changes to your code, we can find the optimal architecture within 5000 samples on Nasbench 101 without manually selecting random seed.
>
> There is no motivation for us to pick random seeds for MLP since it is only a baseline, and we have not manually picked seeds throughout all of our experiments.
>
> We are happy to give you the modified version of your code.

---

> > ### Public Comment · ~Linnan_Wang1 · 2019-10-23
> > **Thank you**
> >
> > "In the real setting, enumerating all models in the search space for prediction seems to be impractical. The previous approaches like Regularized Evolution, MCTS, Random Search, LaNAS, doesn't predict performance on all unseen architectures in the search space.
> >
> > I also did one experiments that show that predicting the performance of every sample is critical for your performance. If you draw 1% architectures from NASBench 101 to predict their performance, the overall performance of MLP quickly deteriorates to 2*10^4 samples to the global optimum. This means that #samples of prediction might contribute substantially to final performance.
> >
> > That being said, more prediction can be very important in reducing the #architectures that are actually being trained, which is a very interesting discovery "
> >
> > See code is in github:https://github.com/linnanwang/MLP-NASBench-101

---

> > > ### Author Response · Authors · 2019-10-29
> > > **The performance of our method is still far better than baselines with your setting**
> > >
> > > We are happy you could replicate our result after modifying your code, we answer your questions in the following:
> > >
> > > Different from one-shot methods, we only store two small matrices (adjacent matrix and feature matrix)  instead of the whole architecture with corresponding weights. Thus, we don't have the similar storage problem as one-shot methods.
> > >
> > > For NASBench dataset, the inference time of predictor for all architectures (420K) is negligible (less than 0.01 seconds). Furthermore, it only occupies 8GB of GPU memory. Thus, it is feasible and logical to fully utilize the prediction power of BOGCN.
> > >
> > > For larger search space, one solution is using sampling methods as we said in algorithm section. Following your suggestion, we randomly sample 1% architectures from the search space for performance prediction. The overall performance of our BOGCN can still find the optimal model with around 2000 samples. After fully training a few hundred architectures, BOGCN could pick out the best architecture as long as it is included in the 1% sample. This experiment further shows the importance of the superior accuracy of BOGCN, other than that, the frequency of sampling also plays an important part in our algorithm. Even though random sampling is good enough, an alternative sampling method can be evolutionary algorithm.
> > >
> > > Another solution is to perform prediction in mini-batches. We have tested our predictor on even larger search space (around half a billion models), the total inference time is around 60 seconds and it consumes only 22GB GPU memory.
> > >
> > > Thanks for your comments, we hope this addresses your concern.

---

> > > > ### Public Comment · ~Linnan_Wang1 · 2019-10-29
> > > > **Thank you for your reply**
> > > >
> > > > Thank you for the detailed explanation, and it is truly impressive to see BOGCN still keeps a similar performance by only predicting 1% random samples. Could you please clarify the performance (i.e. samples to global optimum) of the case using MLP to predict 1% random samples? I really want to know whether this is due to the advantages of GCN representation plus BO or just because there is a discrepancy in the implementation which leads to ~ 10^4 sample complexity in MLP on my side. Thank you.
> > > >
> > > > my implementation: https://github.com/linnanwang/MLP-NASBench-101

---

> > > > > ### Author Response · Authors · 2019-10-30
> > > > > **Your MLP code can be improved but still inferior to BOGCN**
> > > > >
> > > > > Thank you very much for your interest, we share our findings as follows.
> > > > >
> > > > > We have performed the experiment using MLP with your setting, and the result is around 6000 samples to obtain the optimal architecture. We also tried stricter condition (random sample ratio equals 0.1%), and BOGCN can still reach the optimal model with around 4000 samples.
> > > > >
> > > > > If we decrease the random sample ratio (like from 1 to 1%), we can select less proposed sampling models (variable "n" in "propose_location" function in your code). Since BOGCN is accurate enough such that the predicted accuracy of the optimal architecture always belongs to front-rank positions, the shrinking search space means reduced proposed samples every iteration.
> > > > >
> > > > > We hope this answers your question.

---

### Decision · Program_Chairs · 2019-12-19

**Decision:**

Reject

**Comment:**

This paper proposes to use Graph Convolutional Networks (GCNs) in Bayesian optimization for neural architecture search. While the paper title includes multi-objective, this component appears to only be a posthoc evaluation of the Pareto front of networks evaluated using a single-objective search -- this could be performed for any method that evaluates more than one network. Performance on NAS-Bench-101 appears to be very good.

In the private discussion of reviewers and AC, several issues were raised, including whether the approach is compared fairly to LaNAS and whether the GCN will predict well for large search spaces. Also, unfortunately, no code is provided, making it unclear whether the work is reproducible. The reviewers unanimously agreed on a weak rejection score.

I concur with this assessment and therefore recommend rejection.